# Factors Associated with Depression and Anxiety Symptoms among Migrant Population in Spain during the COVID-19 Pandemic

**DOI:** 10.3390/ijerph192315646

**Published:** 2022-11-24

**Authors:** Ivet Bayes-Marin, Maria Roura-Adserias, Iago Giné-Vázquez, Felipe Villalobos, Marta Franch-Roca, Amanda Lloret-Pineda, Aina Gabarrell-Pascuet, Yuelu He, Rachid El Hafi, Fajar Matloob Ahmed Butt, Blanca Mellor-Marsá, M. Carmen Alós, Helena Sainz-Elías, Wala Ayad-Ahmed, Lola Aparicio, Mercedes Espinal Cabeza, Óscar Álvarez Bobo, Yolanda Osorio López, Josep Maria Haro, Paula Cristóbal-Narváez

**Affiliations:** 1Departament de Medicina, Facultat de Medicina i Ciències de la Salut—Campus Clínic, Universitat de Barcelona, 08036 Barcelona, Spain; 2Centre for Biomedical Research on Mental Health (CIBERSAM), 28029 Madrid, Spain; 3Research and Development Unit, Parc Sanitari Sant Joan de Déu, 08950 Barcelona, Spain; 4Fundació Institut Universitari per a la Recerca a l’Atenció Primària de Salut Jordi Gol i Gurina (IDIAPJGol), 08007 Barcelona, Spain; 5Department of Sociology, University of Barcelona, 08036 Barcelona, Spain; 6Departamento de Medicina Legal, Psiquiatría y Patología, Facultad de Medicina, Universidad Complutense de Madrid, 28040 Madrid, Spain; 7Fundación de Investigación Hospital Clínico San Carlos, 28040 Madrid, Spain; 8Servicio de Atención a la Migración en Salud Mental (SATMI), Parc Sanitari Sant Joan de Déu, Avenida Drassanes 19, 08003 Barcelona, Spain; 9Instituto de Investigación Sanitaria Princesa (IP), 28006 Madrid, Spain

**Keywords:** migration, COVID-19, mental health, depression, anxiety

## Abstract

Migrants are likely to experience mental health conditions, being one of the most vulnerable groups during the COVID-19 pandemic. The present study aims to: (1) estimate the prevalence of depressive and anxious symptoms and (2) examine the impact of risk and protective factors on this symptomatology. A sample of 129 migrants living in Spain during the COVID-19 pandemic completed an anonymous online survey, including information on sociodemographic and individual characteristics, migration, basic needs, social environment and perceived health domains. Multiple Poisson regression models analysed the effects of risk and protective factors on depression and anxiety symptoms. The prevalence of depressive and anxiety symptoms was 22.3% and 21.4%, respectively. Risk factors such as living in a rented house and previous mental health conditions were associated with higher depression symptoms, whereas unemployment was related to anxiety symptoms. Conversely, older age, better self-esteem, and higher levels of social support were associated with fewer depression symptoms. Older age and better quality of life were related to fewer anxiety symptoms. These findings addressing risk and protective factors (e.g., social support, self-esteem) help to design culturally effective programs, particularly in migrants with pre-existing mental health conditions, adjusting the organisation of mental healthcare services in difficult times in Spain.

## 1. Introduction

The COVID-19 pandemic has posed an extraordinary health, social and economic challenge to the world. According to the World Health Organisation (WHO), over 400 million people have been infected worldwide, and more than 5 million deaths have been reported since the pandemic’s start [1]. Governments had to implement restrictive policies to control the pandemic, rapidly changing people’s lifestyles [2]. These measures were different in different parts of the world and during phases of the pandemic. In the case of Spain, when the first state of alert was decreed (14 March 2020), strict containment measures were put in place (such as the closure of schools and businesses, the mandatory use of face masks and home confinement). These measures were progressively relaxed in a de-escalation plan that began in May and ended on 20 June 2020, ushering in the “new normality” [3].

Several studies have focused on mental health during the COVID-19 pandemic. A meta-analysis by Wu et al. (2021) showed that the overall pooled prevalence of depression and anxiety was 31.4% and 31.9%, respectively [4]. In the Spanish population, an increase was found in the rate of depression assessed through the two-item Patient Health Questionnaire (PHQ-2), while the levels of anxiety, evaluated by the two-item General Anxiety Disorders Scale, remained stable [5].

Although prior research showed increased levels of symptoms due to the COVID-19 pandemic in the general population [2,5], several studies have also revealed that specific collectives seem to present more pandemic-related symptoms (i.e., religious minorities, women, older people and children) [6]. In this sense, migrants are no exception regarding threats to their mental health, being at a higher risk of developing mental health problems [7,8,9]. Nevertheless, it is interesting to note that migrants were already a vulnerable group before the COVID-19 pandemic [10]. As biopsychosocial theory holds, biological, psychological and social factors contribute to each person’s health or illness experience. In addition to the genetic vulnerability, migrants are more exposed to stressful situations in the migration process, and they also cope with more challenging social scenarios in the new country (i.e., economic problems, housing difficulties, discrimination, prejudices, language barriers, bureaucratic barriers, etc.) [11]. A study conducted by Rousseau et al. (2019) showed that exposure to adverse experiences before, during, or after the migration process and the length of time living in the host country positively correlate with mental health problems [12]. In another study, Bogic et al. (2015) highlighted that migrants with precarious legal status tended to show more stress reactions [13]. Similarly, migrants with low socioeconomic resources, perceived discrimination experiences and interpersonal hassles showed a similar trend [12]. On the contrary, factors such as having a supportive family, educational opportunities, employment, and perceived community integration, seemed to be protective factors against depression and anxiety [12]. It is essential to notice that migrant women were at a higher risk of suffering from depression and anxiety, with more intensity of symptoms than their male counterparts [14]. However, depending on their values and lifestyles, people significantly differ in how they cope with these situations, which can lead to different mental health outcomes [15].

In general, due to social and healthcare implications, immigration is a subject of debate in public health. In the case of Spain, it is particularly relevant because it is a recent phenomenon that has become visible in the last twenty years. In particular, during a period of 8 years (from 2000 to 2008), the registered migrant population grew by more than 4 million. The arrival of foreign immigrants to Spain appeared suddenly and rapidly, and in a short time, figures of such magnitude placed Spain at the same level as other host countries with much greater migratory traditions [16]. In 2021, a total of 457,701 people arrived in Spain, reaching 5,440,148 foreigners living in Spain [17]. The continent where most people come from was Europe (2,164,069), followed by Africa (1,198,573) and South America (1,137,165). In addition, the countries with the highest representation in Spain are Morocco (872,759), Romania (644,473), Colombia (291,751), and Venezuela (199,078) [18].

However, despite growing research on the impact of pandemic-related stressors on mental health, few studies have analysed its effects on migrants’ mental health. For example, Spiritus-Beerden et al. (2021) showed that the mental health of refugees and migrants was affected during the COVID-19 pandemic, especially in some specific groups (i.e., people with insecure housing situations and uncertain residence status, older respondents, and females) who reported higher levels of perceived discrimination and more daily stressors [9] Another study found that those reporting isolation, lower socioeconomic level or lack of access to COVID-19 information were the most vulnerable group to develop psychological distress [19]. Similarly, other studies showed a high prevalence of anxiety and depressive symptoms in migrant workers and a higher vulnerability among those suffering social-related difficulties [20,21]. Nevertheless, it is worth highlighting that much of this research has only focused on specific groups, such as migrant workers or refugees.

To the best of our knowledge, only one previous study examined the mental health of migrants during the COVID-19 pandemic in Spain [22]. This study reported that migrant participants showed worse mental health than non-migrants, and refugees had worse scores on mental health [22]. Nevertheless, in this study, the authors only focused on the role of resilience instead of considering other variables (i.e., met and unmet needs, social environment, and factors related to the migration process) that could also be related to mental health burdens [22]. 

Moreover, most of the cited articles addressing the impact of the COVID-19 pandemic on mental health in migrants were conducted in other countries, so there is a lack of studies conducted in the Spanish context. Since each country has its particularities in terms of restrictions, it is relevant to contextualise the results of each research. For example, the domiciliary confinement implemented in Spain was not required in other countries, and there is evidence that inhabitants of countries with total lockdowns showed worse mental health outcomes [23].

Altogether, research addressing the mental health of the migrant population during the COVID-19 pandemic in the Spanish context is essential for the design and implementation of suitable policies for current or future emergencies. Thus, the present study aims to provide evidence of emotional distress in the migrant population in the context of COVID-19, covering two main objectives: (1) to estimate the prevalence of depressive and anxious symptomatology and (2) to examine the impact of risk and protective factors grouped into six domains (sociodemographic and personal characteristics, migration factors, basic needs, social environment and perceived health) for depressive and or anxious symptoms.

## 2. Materials and Methods

### 2.1. Study Design and Sample Recruitment

Data from a cross-sectional survey—including the migrant population living in Spain during the COVID-19 lockdown—were analysed. An anonymous online survey, performed through the Qualtrics platform [24], was distributed using a convenience and snowball sample using web-based platforms and social media sources. Data were collected from October 2020 to January 2021. The inclusion criteria were: adults aged ≥ 18 years, being migrants and living in Spain during the COVID-19 lockdown (i.e., from March 2020 to June 2020). A total of 129 individuals completed the survey.

Ethical approval was provided by the Fundació Sant Joan de Déu Ethics Committee, Barcelona, Spain (PIC 86-20) and all respondents provided written informed consent before completing the survey (computer-based, e.g., by clicking “yes”) according to the Declaration of Helsinki [25].

### 2.2. Depression and Anxiety (Outcome Variables)

To assess depressive symptoms, we used the 8-item Patient Health Questionnaire (PHQ-8) [26]. Participants were asked to indicate how often they had been bothered by eight possible symptoms in the last 2 weeks (i.e., feeling down, depressed or hopeless, feeling tired or having little energy, etc.), rated 0 “not at all”, 1 “several days”, 2 “more than half the days”, or 3 “nearly every day”. Scores were summed to obtain scale scores ranging from 0 to 24. In the present study, Cronbach’s α coefficient was 0.88.

Anxiety symptoms were evaluated through the 7-item General Anxiety Disorders Scale (GAD-7), a screening measure for generalised anxiety disorder and other anxiety disorders [27]. Participants were asked how often, on a Likert 4-point scale, they experienced several symptoms in the last 2 weeks: feeling nervous, anxious, or on edge, being able to stop or control worrying, worrying too much about different things, trouble relaxing, being restless, becoming easily annoyed or irritable, and feeling afraid as if something awful might happen. Scores were summed, ranging from 0 to 21. Cronbach’s α coefficient was 0.89.

In both measures, higher scores represent worse psychological symptoms. Although we used the continuous variable to perform the regression models, the prevalence of depressive and anxiety symptoms was determined according to the recommended cut-off points for both measures: ≥10 [26,27,28].

### 2.3. CovariatesBased on Previous Research, Covariates Were Classified as Follows [11]

#### 2.3.1. Individual Characteristics

We considered resilience and self-esteem as individual characteristics. In the case of resilience, we used the Connor–Davidson Resilience Scale (CD-RISC) [29], constituted by 10 items, each rated on a 5-point scale (not at all true to true nearly all the time). For our study, we calculated the mean of the scores, where higher scores represent better resilience. To assess self-esteem, we included the Rosenberg Self-esteem Scale (RSS), the most commonly used instrument to measure self-esteem [30]. RSS comprises 10 items related to overall feelings of self-worth or self-acceptance, answered on a 4-point scale ranging from strongly agree to strongly disagree. Since some of them are reversed items, we rescale them into direct ones and sum up the scores, ranging from 0 to 30. Lower scores indicate more self-esteem, and higher scores indicate worse self-esteem. In our study, Cronbach’s α coefficients were 0.89 for the CD-RISC and 0.79 for the RSS.

Moreover, we included other variables in our descriptive analyses to provide an overview of the characteristics of our sample regarding sociodemographic variables and COVID-19 prevalence. Gender was defined as female, male and other (transgender, agender, gender fluid). Age was collected in years as a continuous measure. Furthermore, we considered different levels of education (primary education, secondary education, preparatory education, technical higher education, university higher education) and monthly household income ranges (EUR < 800, EUR 800–1550, EUR 1550–2200, EUR 2200–3600, EUR > 3600). Finally, we provide information regarding COVID-19 diagnosis (including being diagnosed by a doctor or hospitalised due to COVID-19 symptoms).

#### 2.3.2. Factors Related to the Migration Process

In our study, we considered migrants born in a different country from Spain and residing in Spain during the COVID-19 pandemic lockdown, particularly between March 2020 and June 2020. We also included other variables related to the migration process: years since migration (coded as 5 years or less or more than 5 years); country of birth, considering EU citizens (Germany, Bulgaria, France, Italy, Portugal, United Kingdom, and Rumania), non-EU citizens (Argentina, Bolivia, Brazil, China, Colombia, Cuba, Ecuador, Morocco, Pakistan, Peru, Dominican Republic, Venezuela, and Mexico); and residence status, coded as individuals with residence permit (including Spanish citizenship, temporal residence, permanent residence, and student permit) or without a residence permit or in process.

#### 2.3.3. Factors Related to Basic Needs (Housing and Employment)

This section covered two basic needs: housing and employment. Regarding housing, we considered three categories: owned, rented, and other (including living in someone’s house, such as family or a friend). In the case of employment, self-reported current working status, coded as *yes* or *no*, was included.

#### 2.3.4. Factors Related to the Social Environment

We included a modified Oslo Social Support Scale (OSSS-3) to assess the level of social support [31]. This scale consists of only three items focused on the accessibility of practical help. The sum score ranges from 3 to 15; although, we rescaled the items into a 0–12 scale, with higher values representing good levels and low values representing poor levels of social support. In this study, Cronbach’s α coefficient of the modified OSSS-3 was 0.52.

To assess perceived discrimination, we considered the Everyday Discrimination Scale (EDS) [32]. The EDS is a 5-item self-report scale that reflects thoughts and beliefs about experiencing discrimination [33]. Participants were asked to respond 5 items about their perceptions of discrimination: “Are you treated with less courtesy than other people?”, responding to a 6-point Likert-type scale (1 = never, 2 = less than once a year, 3 = a few times a year, 4 = a few times a month, 5 = at least once a week, and 6 = almost every day). In the present study, we created a binary indicator, considering ‘never’ in all the items as ‘no’ and any other answer as ‘yes’. The Cronbach’s α coefficient was 0.70 in our study.

Information about COVID-19-related perceived stress was assessed with an adapted version of the Peri Life Events Scale, a 13-item test [34]. It included several items related to concerns about being infected, loved ones being infected, death of loved ones due to COVID-19, job loss or income reduction due to COVID-19. Each item was rated on a 5-point scale ranging from very intense to none stress. Higher scores indicated lower levels of perceived stress. In the present study, Cronbach’s α coefficient was 0.90.

#### 2.3.5. Perceived Health

Quality of life was evaluated using the EQ-5D-5L, a self-perceived, standardised and generic tool to assess health-related quality of life [35]. The EQ-5D-5L comprises 5 dimensions (mobility, self-care, usual activities, pain/discomfort, anxiety/depression) divided into 5-point response levels (‘no problems’, ‘slight’, ‘moderate’, ‘severe’, and ‘extreme problems). We obtained the scores using the crosswalk value sets for the EQ-5D-5L calculated for Spain, using the methodology developed in a previous work [36], where higher scores mean a better health-related quality of life. The Cronbach’s α coefficient of the EQ-5D-5L based on our study sample was 0.64.

The presence or absence (*yes*/*no*) of non-communicable diseases (NCDs), containing conditions such as respiratory diseases, cardiovascular conditions, diabetes, cancer, chronic liver disease, and immune disorders. Moreover, previous mental health problems (depression, bipolar disorders, panic attacks, anxiety disorders, substance use disorders (drugs or alcohol), and any other mental health problems) were included in the analyses, coded as *yes*/*no*.

### 2.4. Statistical Analyses

The characteristics of the sample were described using summary statistics, with continuous variables being shown as mean (standard deviation) and categories being presented as size (proportion). A multiple Poisson regression model was fitted to analyse the effect of the clusters specified above on migrants having experienced depressive symptoms and anxiety, adjusting by age and gender. Poisson models were preferred to negative binomial estimation as they are more robust except for particular assumptions on the outcomes overdispersion [37]. Estimates were expressed as Incidence Rate Ratios (IRR). In addition, pairwise interactions between COVID-related perceived stress, perceived discrimination, and resilience were investigated. A significance level of 0.05 was adopted. Statistical power for the models was estimated using the R package pwr [38]. All other analyses were performed using the statistical software R version 3.6.3 [39].

## 3. Results

The characteristics of the total sample are presented in Table 1. Our sample was composed of 129 individuals, of which 62.7% were women, and the mean age was 35.3 (SD = 10.7). Regarding socioeconomic characteristics, more than half of the sample had university higher education (56.1%), whereas a small proportion reported having only primary education (1.8%). Conversely, 28.7% reported EUR < 800 household monthly income ranges.

The majority of the sample (76.6%) had lived in Spain for 5 years or less, and 67.5% had a residence permit. In this study, a large proportion of the participants were non-EU citizens (80.0%), especially from Latin-American countries, such as Colombia, Venezuela and Mexico.

Regarding factors related to basic needs, less than half of the sample (44.1%) were working during the recruitment, and 12.9% lived in a non-owned or rented apartment or house. Concerning health, 30.1% of the participants reported some NCDs, 56.3% had previous mental health problems, and a small proportion (10.7%) had a confirmed COVID-19 diagnosis. Finally, according to the cut-off points, the prevalence of depressive and anxiety symptoms was 22.3% and 21.4%, respectively.

Table 2 shows the analyses of risk and protective factors on depressive and anxiety symptoms. Regarding sociodemographic and personal characteristics, increased age was related to a lower risk of depressive symptoms (0.9820 (SE = 0.0050), *p* = 0.0010) and anxiety (0.9790 (SE = 0.0060), *p* = 0.0000) symptoms. In contrast, better self-esteem appeared to be protective only for depressive symptoms (1.0570 (SE = 0.0110), *p* = 0.000). With regard to migration variables, being an EU citizen was a risk factor for both outcomes, depressive symptoms: (1.6880 (0.3320), *p* = 0.0080); anxiety symptoms: (1.6550 (0.3420), *p* = 0.0150). Furthermore, among unmet basic needs, unemployment (1.3840 (SE = 0.1400), *p* = 0.0010) was a risk factor for anxiety symptoms, while living in a rented house and other housings were significant risk factors (1.4890 (SE = 0.2790), *p* = 0.0340; 1.7830 (SE = 0.3760), *p* = 0.0060).

Regarding social environment variables, social support was a significant protective factor for depressive symptoms (0.9280 (SE = 0.0210), *p* = 0.0010) but not for anxiety symptoms (0.9930 (SE = 0.0230), *p* = 0.7780). Finally, in terms of perceived health, our results showed that a better quality of life was linked to a lower risk of anxiety symptoms (0.5030 (SE = 0.1290), *p* = 0.0070), and previous mental health problems were related to a significantly increased risk for depressive symptoms (1.2980 (SE = 0.1450), *p* = 0.0190). 

To confirm previous results and acknowledge that the sample size was small for the large set of predictors of the main models, we also analysed block models with a smaller number of predictors belonging to each cluster and adjusted by gender and age (see Appendix A). These block models shared positive statistical significance with the trends of the effects reported in the general model (Table 2), except for the housing effects.

## 4. Discussion

After reviewing previous research on mental health in the migrant general population during the COVID-19 pandemic, we spotted a gap in the literature. As we mentioned earlier, other studies focused on specific people, such as refugees or migrant workers [9,16,17,18] and have been primarily conducted in other countries. Nevertheless, evidence-based knowledge about migrant general population needs and mental health problems in this understudied context is required to provide guidelines and resources to cope with actual and future emergencies [40]. Thus, the present study aimed to estimate the prevalence of depressive and anxious symptomatology, and to examine risk or protective factors for depressive and/or anxious symptoms in a sample of 129 migrant people who had been living in Spain during the COVID-19 lockdown.

In general terms, our sample is mainly constituted of women (62.7%), young participants (mean of 35.3 years old), and people who reported having a higher university education (56.1%). However, 28.7% declared to be in the EUR < 800 monthly income range, and almost 60.0% of the individuals were not working during the recruitment period. This population profile is not uncommon in the migration context. It is known that migrant professionals show higher job insecurity, reflected in lower salaries and difficulties in obtaining an appropriate job, hampered by the low-paced recognition of foreign education and qualifications [40]. Regarding the migratory status, the participants who enrolled in our study were predominantly people who had lived for 5 years or less in Spain (76.6%), and 67.5% reported having residence permits. Concerning the prevalence of depressive and anxiety symptoms (22.3% and 21.4%, respectively), our results are in line with previous research conducted with the non-migrant general population [41]. Valiente et al. (2021) using the same assessment tools (PHQ-9 and GAD-7) and cut-offs, found similar rates of depression (22.1%, 95% (20.1, 24.0%)) and anxiety (19.6%, 95% (17.8, 21.6%)) [41]. Notwithstanding this, the available systematic reviews about psychological distress in the COVID-19 era suggest a huge heterogeneity in the results [2,4]. Thus, estimating the depressive and anxiety symptoms rate in this context can be cumbersome since several factors may act synergistically.

We identified different risk and protective factors for depressive and anxiety symptoms, which were analysed considering several blocks of information to focus on the differential impact of these mental health symptoms. Regarding sociodemographic and personal characteristics, older age was a protective factor for depressive and anxiety symptoms. This result is in line with what has been found in the non-migrant population in the COVID-19 context [42,43]. A cross-sectional study in China found that younger individuals reported a significantly higher prevalence of generalised anxiety disorder and depressive symptoms than older people [42]. In the same line, other authors spotted that the anxiety risk of people above 40 years old was 0.40 times more (95% CI 0.16–0.99) than those below 40 years old [43]. One possible explanation may be greater exposure to general stressors in younger individuals, such as virus jeopardy in work and social environments [42,44].

Higher self-esteem has been consistently reported as a protective factor for mental health [45]. More specifically, much research highlights the association between low self-esteem and depressive symptoms [46,47]. According to the vulnerability model, low self-esteem contributes to the development and maintenance of depressive symptoms through both intrapersonal and interpersonal mechanisms [46]. In the migration context, self-esteem becomes especially relevant, as multiple factors might be involved (i.e., reassurance-seeking about personal worth, social network, socioeconomic status, experiences of discrimination, etc.), acting as a resilient component buffering the impact of adverse events on mental health. In the pandemic context, self-esteem also became especially relevant, as was reported in previous studies [48,49]. We also included a resilience measure in the analyses, although it did not reach statistical significance. Future studies should include other proxy measures of resilience, such as coping strategies, post-traumatic growth, or even other measures of subjective well-being (i.e., purpose in life, life meaning), to better capture static and dynamic elements of resilience.

According to our results, EU citizens had a higher risk for both symptoms concerning migration variables. Furthermore, a study using data from 21 countries from the European Social Survey and the Greek MIGHEAL survey showed high levels of depressive symptoms among migrant and non-migrant groups in Eastern and Mediterranean European countries [50]. These findings resonate with our results. EU citizens were originally from France, Italy, and Romania, followed by the regions mentioned above where higher levels of depressive symptoms were identified. Nevertheless, our sample’s small proportion of EU citizens (*n* = 5) might limit our ability to draw firm conclusions.

Regarding basic needs, not having employment was a risk factor for having anxiety symptoms, and living in a rented house or other housing conditions (i.e., living in a family or a friend’s house) were risk factors for depressive symptoms. Regarding unemployment, migrants were often over-represented in jobs hit harder by the pandemic, such as food services and domestic work [40]. In addition, they are often without unemployment benefits, paid sick leave, or even days off, which might increase the risk for anxiety symptomatology [40]. Concerning housing conditions, rented or overcrowded shared flat made it challenging to respect social distancing or quarantine in case of COVID-19-related symptoms [40]. However, living in a family or a friend’s house could be a consequence of job loss, and, in turn, it could be associated with depressive symptoms due to feelings of ineffectiveness.

In addition, our results showed that perceived social support was a significant protective factor for depressive symptoms. Several studies concluded that perceived social support was strongly positively associated with mental health [51]. This is particularly relevant in a context of crisis, characterised by forced isolation, uncertainty and several losses (e.g., employment, housing and loved ones). A study framed in the COVID-19 context found that resilience had a positive association with mental health, and social support served as a buffer against the negative impact of low levels of resilience on mental health. Importantly, it was found that this effect did not vary among age groups [52].

Finally, regarding health-related variables, a better quality of life seemed to be a protective factor for anxiety. Previous studies pointed out the association between quality of life and anxiety [53,54]. In addition, the quality-of-life assessment often includes functioning evaluation, as in the EQ-5D-5L [55]. These factors could be acting synergistically so that being functionally impaired (reporting worse quality of life) could affect mental health through a worse self-concept and self-esteem. Being anxious or depressed because of the symptoms would lead to impaired quality of life and functionality. Moreover, one of the dimensions of the EQ-5D-5L is anxiety/depression, which in some way supports our interpretation. Contrarily, having a previous mental health diagnosis was a risk factor for depressive symptoms. This finding has been widely supported in the COVID-19-related literature, suggesting a greater vulnerability in those with previous mental health problems, leading to a recurrence of psychological symptoms [44,56]. Since migrants are at-risk populations, whose mental health is often neglected [8], more actions should be taken to improve their global well-being.

## 5. Strengths and Limitations

The present work’s strengths include analysing a sample of migrants from the general population rather than focusing only on specific population groups (i.e., migrant workers and refugees). Moreover, studying depressive and anxiety symptoms separately allowed us to identify differentiating factors that may guide suitable interventions for each mental health condition. To this end, we included in the analysis several variables that, according to the literature, can impact mental health in the migrant population. In addition, the inclusion of extensively validated instruments in the study shed light on those factors involved in improved or worsened mental health.

However, some limitations are to be considered when interpreting the results. First, the survey was available only in Spanish. This might have introduced some bias since only individuals from Latin American countries or migrants living in Spain for enough time to learn the language could participate in the study. Both factors could impact the representativeness of our results. On the one hand, cultural issues may be related to a differential impact of the studied variables on mental health.

On the other hand, the time spent in the host country is a relevant feature linked with mental health outcomes [8]. Nevertheless, the sample mainly consists of people from Latin America (80.0%), which implies a greater homogeneity. This is important because it was a relevant result; the cultural distance between Latin American countries and Spain is smaller than with other cultures [57]. Despite this, we found poor mental health outcomes in our sample. This may suggest that people from more distant cultures living in Spain who usually have to cope with more challenges may present more mental health problems in a crisis and, thus, a greater vulnerability that demands targeted interventions. Second, our results are part of a cross-sectional online survey. Because of this, individuals with greater exposure or without Internet access could not enrol in the study, which may impact the representability of the results. In addition, the inherent nature of cross-sectional studies does not allow for the drawing of causality conclusions, so future longitudinal research should be conducted to understand these associations’ directionality better. Future research should consider culturally adapting surveys than can allow higher participation and better representativity of the sample of diverse cultural contexts. In addition, instead of an online survey, a face-to-face interview may lessen sociocultural barriers, such as problems with Internet access, technological barriers or difficulties in understanding some of the items.

## 6. Conclusions

Migrants face a unique set of challenges that impact their mental health status, constituting a public health concern. In this sense, there is an urgent need for a clearer understanding of mental health-related factors during and after the COVID-19 outbreak in the migrant community. In our study, we identified some variables associated with worse mental health outcomes related to unmet basic needs, such as unemployment and living in a rented or shared flat and predisposing factors, such as previous mental health problems. However, notwithstanding the above-mentioned, we also found protective factors for depressive and anxiety symptoms, namely, better self-esteem, quality of life and perceived social support. These findings can help to improve policies and programs addressed to improve mental health in people that have migrated to Spain and design culturally effective strategies, adjusting the organisation of mental healthcare services.

## Figures and Tables

**Table 1 ijerph-19-15646-t001:** Characteristics of the sample.

Variables	Total Sample (*n* = 129)
Sociodemographic and individual characteristics	
Gender, *n* (%)	
Male	44 (34.9)
Female	79 (62.7)
Other	3 (2.4)
Age, mean (SD)	35.3 (10.7)
Level of education, *n* (%)	
Primary education	2 (1.8)
Secondary education	18 (15.8)
Preparatory education	5 (4.4)
Technical higher education	22 (19.3)
University higher education	64 (56.1)
Household monthly income ranges (EUR), *n* (%)	
<800	29 (28.7)
800–1550	29 (28.7)
1550–2200	8 (8.0)
2200–3600	7 (6.9)
>3600	3 (3.0)
Resilience, mean (SD)	2.9 (0.8) *
Self-esteem, mean (SD)	8.9 (4.9) *
Migration	
Years since migration (≤5), *n* (%)	82 (76.6)
Residence permit (yes), *n* (%)	79 (67.5)
Country of birth, *n* (%)	
EU citizens	5 (4.0)
non-EU citizens	100 (80.0)
Basic needs	
Employment (yes), *n* (%)	45 (44.1)
Housing, *n* (%)	
Owned	9 (8.9)
Rented	79 (78.2)
Other	13 (12.9)
Social environment	
Social support, mean (SD)	5.6 (2.1) *
COVID-related perceived stress, mean (SD)	1.9 (0.9) *
Discrimination (yes), *n*(%)	88 (94.6)
Perceived health	
Quality of life, mean (SD)	0.9 (0.2) *
NCDs diagnosis (yes), *n* (%)	28 (30.1)
Previous mental health problems (yes), *n* (%)	58 (56.3)
Confirmed COVID-19 diagnosis, *n* (%)	12 (10.7)
Depressive symptoms (yes), *n* (%)	23 (22.3) *
Anxiety symptoms (yes), *n* (%)	22 (21.4) *

* In the present study, the minimum and maximum scores for each scale were: resilience (CD-RISC: 0–4), self-esteem (RSS: 0–21), social support (OSSS-3: 0–10), COVID-related perceived stress (Peri Life Events Scale: 0–4), quality of life (EQ-5D-5L: 0–1), depressive symptoms (PHQ-8: 0–24), and anxiety symptoms (GAD-7: 0–19).

**Table 2 ijerph-19-15646-t002:** Results from Poisson regression analyses with depressive symptoms and anxiety as outcome variables in the total sample.

Variables	Models					
	Depressive Symptoms			Anxiety Symptoms		
	IRR	SE	*p*-Value	IRR	SE	*p*-Value
Sociodemographic and individual characteristics						
Gender						
Male (ref.)	-	-	-	-	-	-
Female	1.0560	0.1050	0.5820	0.9800	0.0990	0.8410
Other	0.0000	0.0000	0.9940	0.0000	0.0000	0.9840
Age	0.9820	0.0050	0.0010	0.9790	0.0060	0.0000
Resilience	0.0000	0.0540	0.9870	0.2270	0.2370	0.1570
Self-esteem	1.0570	0.0110	0.0000	1.0140	0.0110	0.1970
Migration						
Residence permit						
No (ref.)	-	-	-	-	-	-
Yes	0.9830	0.1130	0.8790	1.1680	0.1390	0.1930
Country of birth						
Non-EU citizens (ref.)	-	-	-	-	-	-
EU citizens	1.6880	0.3320	0.0080	1.6550	0.3420	0.0150
Basic needs						
Employment						
Yes (ref.)	-	-	-	-	-	-
No	1.0380	0.0980	0.6940	1.3840	0.1400	0.0010
Housing						
Owned (ref.)	-	-	-	-	-	-
Rented housing	1.4890	0.2790	0.0340	0.9010	0.1500	0.5280
Other	1.7830	0.3760	0.0060	0.9230	0.1840	0.6880
Social environment						
Social support	0.9280	0.0210	0.0010	0.9930	0.0230	0.7780
COVID-related perceived stress	0.0030	1.3760	0.9890	0.4670	0.2200	0.1060
Discrimination	0.0000	0.0000	0.9870	0.0240	0.0510	0.0800
Perceived health						
Quality of life	0.8760	0.2230	0.6040	0.5030	0.1290	0.0070
NCDs diagnosis						
No (ref.)	-	-	-	-	-	-
Yes	1.0750	0.1040	0.4580	0.9720	0.0970	0.7770
Previous mental health problems						
No (ref.)	-	-	-	-	-	-
Yes	1.2980	0.1450	0.0190	0.9760	0.1120	0.8350

Note. Models were adjusted for gender and age. The goodness of fit of each model (depressive and anxiety symptoms, respectively) was estimated through the following indices: Akaike Information Criterion (AIC: 553.2, 541.5), Bayesian Information Criterion (BIC: 606.4, 594.7), pseudo-R2 (0.498, 0.460), and adjusted R2 (0.364, 0.314). The models reached the adjusted R2 with 99.5% and 97.5% statistical power at a 0.05 significance level.

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
