# Peer review of "Factors Associated with Depression and Anxiety Symptoms among Migrant Population in Spain during the COVID-19 Pandemic"

_ijerph, 2022, doi:10.3390/ijerph192315646_

Round 1
Reviewer 1 Report
I am pleased of reviewing the manuscript “Factors Associated with Depression and Anxiety Symptoms 2 Among Migrant Population in Spain During the COVID-19 3 Pandemic” (Manuscript No. ijerph-1950049), submitted to the International Journal of Environmental Research and Public Health.
Summary
The paper addresses the mental health status of Spanish immigrants during the COVID-19 pandemic. In order to do so, they collect a convenience and snowball sample of migrants through Qualtrics (applying the PHQ-8 and the GAD-7 to study depression and anxiety, respectively) which they analyse making use of Poisson regression models. Their results suggest the relevance of the housing tenure status, previous mental health conditions and unemployment.
Contribution and overall assessment
The main contribution of the authors consists in the fact that the paper is the first study that explores mental health issues among migrants in Spain. On the one hand, the sampling procedure allows them to perform a pioneering work on this topic and to collect detailed information on mental health outcomes that would not be available otherwise in Spain. Overall, the quantitative analysis is overall right, employing a count model to explore the main determinants of the outcomes. On the other hand, the sample is not large and prevent drawing strong inference on some issues and the sampling procedure is not exempt from problems. Addressing this topic, in the middle of turmoil experience by Spain during the period of analysis is complex. Overall, weighting the pros and cons mentioned above, I think that it is worthy to carry out this sort of study, even with imperfections given the remarkable absence of good data for Spain. Nevertheless, I think that the authors should address several points before the paper can be published in an academic journal. I describe them below.
Minor comments
First, I think that non-Spanish readers would benefit from some context for immigration in Spain. This could be done either in the introduction or in the results section (when talking about the composition of the sample). It could be done either in the main text or in a footnote. From my point of view, it would be necessary that the authors indicate, including several references, that, in contrast to other high-income countries, migration is a relatively recent phenomenon in Spain, particularly, since the mid-1990s (Bover & Velilla, 2005) and that, among non-EU citizens, Latin American and Caribbean migrants represent the largest group (Muñoz de Bustillo & Antón, 2010). I suggest including a paragraph or a couple of sentences with the mentioned references.
Second, I think that it could be a good idea to link the variable length of residence in Spain to previous studies that show that mental health of migrants worsens with time of residence (Rivera et al., 2015; Ronda et al., 2019).
Third, I think that the quantitative analysis is competently done. Nevertheless, some readers might question the use of Poisson regression model instead of negative binomial regression models I suggest highlighting that Poisson model requires weaker assumptions for providing consistent estimates (see, e.g., Blackburn [2014] or Wooldridge []).
Fourth, the quantitative analysis does not seem to present any measure of goodness of fit. I suggest presenting either some Bayesian measure (AIC or BIC) or the pseudo-R2 proposed by Zheng and Agresti (2000).
Last, overall, formal issues are adequately taken into account by the authors. Particularly, the organization of the manuscript is good. Nevertheless, several minor points deserve some additional attention. For instance, the authors should use the Present Simple to refer to their study when possible (e.g., the aims of the study). In lines 113–115, they wrongly use the Past Simple (“this study aimed”).
References
Blackburn, M. L. (2014). The relative performance of Poisson and negative binomial regression estimators. Oxford Bulletin of Economics and Statistics, 77(4), 605-616. https://doi.org/10.1111/obes.12074
Bover, O., & Velilla, P. (2005). Migrations in Spain: historical background and current trends. In K. F. Zimmermann (Ed.), European migration: what do we know? (pp. 389–414). Centre for Economic Policy Research.
Muñoz de Bustillo, R., & Antón, J.-I. (2010). De la España que emigra a la España que acoge: contexto, dimensión y características de la inmigración latinoamericana en España. América Latina Hoy, 55, 15–39. https://doi.org/10.14201/alh.7261
Rivera, B., Casal, B., & Currais, L. (2015). Length of stay and mental health of the immigrant population in Spain: evidence of the healthy immigrant effect. Applied Economics, 47(19), 1972–1982.
Ronda, E., Martínez, J. M., Reid, A., & Agudelo-Suárez, A. A. (2019). Longer Residence of Ecuadorian and Colombian Migrant Workers in Spain Associated with New Episodes of Common Mental Disorders. International Journal of Environmental Research and Public Health, 16(11), 2027. https://doi.org/10.3390/ijerph16112027
Wooldridge, J. M. (2010). Econometric analysis of cross section and panel data (2nd ed.). The MIT Press.
Zheng, B., & Agresti, A. (2000). Summarizing the predictive power of a generalized linear model. Statistics in Medicine, 19(13), 1771–1781. https://doi.org/10.1002/1097-0258(20000715)19:13%3C1771::AID-SIM485%3E3.0.CO;2-P
Reviewer 2 Report
Thank you for the opportunity to review this manuscript: "Factors Associated with Depression and Anxiety Symptoms Among Migrant Population in Spain During the COVID-19 3 Pandemic". I find the topic to be of overall interest, yet I note several difficulties with the manuscript. There are strengths. The paper proposes an interesting exploration of the mental health of the migrant population during the pandemic. The article is well-written and addresses an important topic. The abstract is good and contains important elements (with one reservation). The article's good side is also presenting the results, tables, structure, readability, and sound scientific language.
However, a significant number of concerns limit my overall enthusiasm. Some of the most serious concerns about this manuscript are noted below.
- The authors did not make a strong case for examining their chosen variables in the present study. The theoretical background is not enough. The relationship between depression and anxiety does not have good ground in the introduction. In this study, psychological variables are considered; therefore, the biggest drawback is the lack of a psychological explanation of the investigated mechanisms.
- The study looks like a listing of different aspects with little or no connection to each other. The introduction does not justify the selection of the variables constituting the predictors in the model. It was also not syncretically included in the discussion. Researchers are characterized by high analyticity, and this is a great advantage. However, the discussion of the results requires a certain level of generalization.
- The most important concern relates to the size of the study group and statistical complexity. Researchers must answer the question about the size of the studied sample. What justifies its size? Does the adopted model of statistical analysis allow the inclusion of such a large number of variables with a relatively small sample? It seems that the study group must be larger, or the model should be simplified with the current group size (N = 129 ). In my opinion, the results should be recalculated. It seems to be a reasonable proposition given the group's size and the number of analyzed relations.
Researchers divide independent variables into individual categories in a very clear way. However, each of these categories has its representations in two, three, or more variables, and for most of them, standardized research tools are used. It would be necessary to calculate how many total variables were put into the model and whether it is valid with N equal to 129. Perhaps a choice should be made between the variables - referring to psychological or sociological theories.
Other comments
Abstract
Line 37-39
"Future studies are needed to better understand migrants' mental health during and after the COVID-19 outbreak, to create mental health programs to reduce health disparities in vulnerable groups"
Perhaps more important in the abstract will be the sentence synthesizing the obtained results and interpreting them.
Line 54-58
"Several studies have focused on mental health during the COVID-19 pandemic. A meta-analysis by Wu et al. (2021) showed that the overall pooled prevalence of depression and anxiety was 31.4% and 31.9%, respectively [4]. In the Spanish population was found an increase in the rate of depression assessing through the 2 -item Patient Health Questionnaire (PHQ-2) (B = 0.31, p < 0.01), while the levels of anxiety, evaluated by the 2-item General Anxiety Disorders Scale, remained stable (GAD-2: B = −0.014, p = 0.752) [5]."
The statistical results seem unnecessary in the introduction.
Line 90
We can read this: Haga clic o pulse aquí para escribir texto.
Line 113
The aim of the study
The aim of the research requires reformulation. There is no clear definition of what the aim is. What are the researchers' hypotheses? From what they come from. Starting with line 97, researchers present a study they appreciate while pointing to shortcomings. Aspects that did not appear in that described study later appear in the presented results. It must be clearly stated that these are the areas that constitute the aim of the research presented in the article.
Line 190-195
Line 190-197
"To assess perceived discrimination, we considered the Everyday Discrimination Scale (EDS) [29]. The EDS is a 5-item self-report scale that reflects thoughts and beliefs about experiencing discrimination [30]. Participants were asked to respond 5 items about their perceptions of discrimination: "Are you treated with less courtesy than other people?", responding to a 6-point Likert-type scale (1=never, 2=less than once a year, 3=a few times a year, 4=a few times a month, 5=at least once a week, and 6=almost every day). In the present study, we created a binary indicator, considering 'never' in all the items as 'no' 196 and any other answer as 'yes'."
It is unclear why researchers are changing a continuous variable to a nominal one.
Generally
I do not see a measure of internal consistency for any of the measures used. So, I kindly ask the authors to add a measure of internal consistency (e.g., Cronbach's alpha) for all the measures.
It also provides information on the minimum and maximum results on all scales. Without it is not easy to find a reference point when analyzing the results of the studied group.
Discussion line275-284
This is not a discussion but a description of the results and should probably be included in the results section.
Line 284-287
"Concerning the prevalence of depressive and anxiety symptoms (22.3% and 21.4%, respectively), our results were in line with previous research conducted with the non-migrant general population [36]. Valiente et al. (2021) using the same assessment tools (PHQ-9 and GAD-7) and cut-offs, found similar rates of depression (22.1%, 95% [20.1, 24.0%]) and anxiety (19.6%, 95% [17.8, 21.6%]) [36].”
Estimating the depressive and anxiety symptoms rate was not the study's aim.
Line 314-315
"We also included a resilience measure in the analyses, although it did not reach statistical significance, probably due to the small sample size."
This explanation does not seem to be good, the group was not too small for other variables (such as self-esteem), and it was too small for the resilience variable.
Line 355
"Moreover, one of the dimensions of the EQ-5D-5L is anxiety/depression, which in some way supports our interpretation"
This is the main problem resulting from the adopted research model. It has no theoretical support, and ultimately the same is explained by the same.
Although the article is generally well-written and deserves to be published in this journal, some necessary revisions must be made before the article's publication.
